# Stochastic representation of many-body quantum states

Hristiana Atanasova [1], Liam Bernheimer [1] & Guy Cohen [1,2] ✉

The quantum many-body problem is ultimately a curse of dimensionality: the state of a system with many particles is determined by a function with many dimensions, which rapidly becomes difficult to efficiently store, evaluate and manipulate numerically. On the other hand, modern machine learning models like deep neural networks can express highly correlated functions in extremely large-dimensional spaces, including those describing quantum mechanical problems. We show that if one represents wavefunctions as a stochastically generated set of sample points, the problem of finding ground states can be reduced to one where the most technically challenging step is that of performing regression—a standard supervised learning task. In the stochastic representation the (anti)symmetric property of fermionic/bosonic wavefunction can be used for data augmentation and learned rather than explicitly enforced. We further demonstrate that propagation of an ansatz towards the ground state can then be performed in a more robust and computationally scalable fashion than traditional variational approaches allow.

The state of a quantum system is encoded in the wavefunction, a high-dimensional object that cannot generally be represented or manipulated using a classical computer. Yet, the challenge of experimentally characterizing and theoretically predicting the ground states of many-body systems remains central in the physical sciences[1-5]. Historically, some of the most successful computational approaches have relied on variational principles[6]. One assumes an approximate, parameterized functional form for the $n$-body wavefunction in terms of a set of parameters $\vartheta$: that is, $\Psi(\mathbf{r}_1, \ldots, \mathbf{r}_n) \simeq \Phi_\vartheta(\mathbf{r}_1, \ldots, \mathbf{r}_n)$. Given this ansatz, in order to find the ground state one then attempts to minimize $\varepsilon(\vartheta) \equiv \frac{\langle \Phi_\vartheta | \hat{H} | \Phi_\vartheta \rangle}{\langle \Phi_\vartheta | \Phi_\vartheta \rangle}$ with respect to $\vartheta$.

Early variational approaches to quantum mechanics, like the pioneering work of Hylleraas on the Helium atom[7], date back almost a century. These methods rely on very simple ansatzes that generate analytical expressions for the energy, and their extensions[8] remain the most accurate algorithms to date for this problem[9]. Later approaches include variational Monte Carlo (VMC) techniques that allow much more general functional forms by evaluating and optimizing $\varepsilon(\vartheta)$ stochastically[10-14]. In a recent surge of activity, modern deep artificial neural networks (NNs) and other machine learning (ML) models[15,16] have

emerged at the forefront of such studies[17]. These ideas have been applied to both phenomenological spin/lattice models[18-25], and real-space fermionic or bosonic models used in, e.g., quantum chemistry and nuclear physics[26-32]. The rapid stream of advancements has culminated in impressive and technically sophisticated new approaches that are already competitive with established methods in some ways[33-37].

A variety of restrictions have so far been placed on the ansatzes. Perhaps most notably, with respect to the real-space models, a small number of permanents or determinants has typically been used to express the wavefunction. This explicitly enforces the symmetry or antisymmetry characterizing wavefunctions of identical bosons and fermions, respectively. It also enables taking full advantage of mean-field solutions as a starting point[22,33,34]. However, the universal nature of deep neural networks as estimators for correlated multidimensional functions[38], together with their successful employment in a variety of seemingly disparate fields[15], suggests that dramatic improvements may be in reach if unrestricted networks could be used directly. This would also enable the use of promising modern probabilistic models like autoregressive networks[23,39]. Furthermore, most quantum chemistry methods rely on representing electronic wavefunctions as linear combinations of determinants. Since the number of relevant determinants

[1]School of Chemistry, Tel Aviv University, Tel Aviv 6997801, Israel. [2]The Raymond and Beverley Sackler Center for Computational Molecular and Materials Science, Tel Aviv University, Tel Aviv 6997801, Israel. ✉e-mail: gcohen@tau.ac.il

grows exponentially with system size and the cost of evaluating each determinant grows cubically, the computational cost of high-accuracy methods increases rapidly with the number of electrons, though recent models have been able to circumvent at least some of these scaling issues[33,34]. Finally, optimization has often presented difficulties. This led most researchers to employ relatively expensive optimization schemes like the stochastic reconfiguration/natural gradient descent method[40–42], which are more expensive than normal gradient descent to scale to models with a large number of parameters. Recent work has made great progress in exceeding such limitations[43–46], but they remain important.

In some cases, the Schrödinger equation can be recast in stochastic form. For example, in diffusion Monte Carlo (DMC)[14,47–49], quantum properties can be extracted from the simulated stochastic dynamics of a set of $N$ walkers $\{\mathbf{R}_i\}$, with $i \in \{1, \ldots, N\}$ denoting a walker index and $\mathbf{R}_i \equiv \mathbf{r}_{1,i} \oplus \ldots \oplus \mathbf{r}_{n,i}$ containing the coordinates of all $n$ particles in walker $i$. It is never necessary to store or evaluate wavefunctions explicitly. However, the DMC algorithm breaks down when the wavefunction has nodal surfaces: it exhibits a sign problem[47,50], meaning that the variance of the stochastic estimator (and therefore the error) grows exponentially with the system's size.. The errors due to the sign problem can only be controlled when an accurate and efficiently evaluated expression for the location of the nodal surfaces is known[48,51]. This suggests that one could improve the precision of a DMC calculation by preceding it with a variational calculation, which would then be used to estimate the location of the nodes. The overall accuracy is then limited by that of the variational ansatz $\Phi_\vartheta$ and the optimization procedure. In practice, therefore, DMC is most often employed as the final step of a VMC calculation, providing a small correction and enhancing the accuracy. This is not a problem specific to DMC: in fermionic systems, sign problems generically afflict Monte Carlo methods that circumvent the need to enumerate wavefunctions[52].

Here, we propose an intermediate approach that allows for full utilization of the expressive power of advanced ML models, without sacrificing either computational scaling or systematic optimization. Importantly, in our method the (anti)symmetry of fermions and bosons becomes an advantage rather than a drawback; and the sign problem is controlled.

## Results

### Stochastic representation of the wavefunction

Here, we present the main idea of this work: stochastic representation of wavefunctions. Instead of representing the wavefunction by a density of walkers $\{\mathbf{R}_i\}$, wavefunction samples $\left\{\left(\mathbf{R}_i^{(j)}, \Psi_s^{(j)}(\mathbf{R}_i)\right)\right\}$ are used as our representation in iteration $j$. The steps are as follows:

1. Obtain samples $\{(\mathbf{R}_i^{(0)}, \Psi_s^{(0)}(\mathbf{R}_i))\}$; e.g., from a non-interacting, perturbative or VMC/DMC calculation, and set $j = 1$.
2. Perform stochastic projection (defined below) of samples $\{(\mathbf{R}_i^{(j-1)}, \Psi_s^{(j-1)}(\mathbf{R}_i))\}$ onto the symmetric or antisymmetric subspace, represented respectively by the operator $\hat{P}_{S/A}$.
3. Perform regression: use the projected samples $\{(\mathbf{R}_i^{(j-1)}, \Psi_s^{(j-1)}(\mathbf{R}_i))\}$ to train an ML model expressing a projected continuous trial function $\hat{P}_{S/A}\Phi_\vartheta^{(j-1)}(\mathbf{r}_1, \ldots, \mathbf{r}_n)$.
4. Given the trial function, generate a new set of sample coordinates $\{\mathbf{R}_i^{(j)}\}$, and perform imaginary time propagation over interval $\Delta\tau$ on the wavefunction at the sample coordinates with respect to the Hamiltonian $\hat{H}$.
5. Repeat steps 2–4 until converged. Steps 2–4 can also be expressed in the succinct form:

$$\Psi_s^{(j)}\left(\mathbf{R}_i^{(j)}\right) = e^{-\Delta\tau\hat{H}}\hat{P}_{S/A}\Phi_\vartheta^{(j-1)}(\mathbf{R})\big|_{\mathbf{R}=\mathbf{R}_i^{(j)}}. \tag{1}$$

They can be repeated as many times as needed to find the closest possible approximation to the ground state given the ansatz, which (up to a normalization factor) is given by $\lim_{n\to\infty}\left(e^{-\Delta\tau\hat{H}}\hat{P}_{S/A}\right)^n\Phi_\vartheta(\mathbf{R}_i)$.

The procedure outlined above and described in greater detail below is related and complementary to both VMC and DMC, but is clearly distinct from both. For example, the samples need not be distributed with respect to $|\Psi(\mathbf{R}_i)|^2$, as walkers in DMC do—though, it is usually helpful for the sake of importance sampling to have at least part of them distributed thus. The machine learning optimization procedure includes no reference to the variational energy $\varepsilon(\vartheta)$ or its gradient $\nabla_\vartheta\varepsilon(\vartheta)$, as in VMC. The energy is calculated from $\Phi_\vartheta(\mathbf{R})$ only if and when it is desired as an observable, using standard Monte Carlo techniques. Instead of being set for the entire calculation as in VMC, the ansatz $\Phi_\vartheta(\mathbf{R})$ can be replaced by a completely different parametrization as many times as desired between time propagation steps. This can be very useful when starting with an ansatz that greatly differs from the ground state wavefunction the algorithm will eventually converge to. The network required to obtain a good fit of the wavefunction, the learning rate and the number of samples can all change as the wavefunction evolves.

### Stochastic projection

Generically, a spatial wavefunction with some stochastic component that undergoes imaginary time evolution will go to its bosonic ground state, which is symmetric to particle exchanges. For the algorithm to be useful in electronic problems, it is crucial that we be able to target both symmetric/bosonic states and antisymmetric/fermionic states. A very useful property of the stochastic representation is that it is possible to project out the undesired components by directly acting on the data. This obviates the need for explicitly enforcing symmetry conditions within the ansatz. Alternatively, symmetric ansatzes can still be used instead.

The main idea is that, given the set of samples $\{(\mathbf{R}_i, \Psi_s(\mathbf{R}_i))\}$, one can—at negligible expense—take advantage of the exchange symmetry/antisymmetry to generate some or all of the new samples

$$\left\{\left(P\mathbf{R}_i, (\pm1)^{\text{sign}(P)}\Psi_s(P\mathbf{R}_i)\right)\right\}. \tag{2}$$

Here, $P$ is one of the $n!$ possible permutation operators that can act on the $n$ single-particle coordinates $\mathbf{r}_{1,i}, \ldots, \mathbf{r}_{n,i}$ composing $\mathbf{R}_i$; sign$(P)$ is its parity; and the value $\pm1$ is used for bosons and fermions, respectively. For example, in a fermionic three particle system and for $P = (3 \quad 2 \quad 1)$, sample $(\mathbf{r}_{1,i} \oplus \mathbf{r}_{2,i} \oplus \mathbf{r}_{3,i}, \Psi_s(\mathbf{r}_{1,i}, \mathbf{r}_{2,i}, \mathbf{r}_{3,i}))$ produces the new sample $(\mathbf{r}_{3,i} \oplus \mathbf{r}_{2,i} \oplus \mathbf{r}_{1,i}, -\Psi_s(\mathbf{r}_{1,i}, \mathbf{r}_{2,i}, \mathbf{r}_{3,i}))$. Since the number of possible new samples grows factorially with the number of particles in the system, it will generally be better to generate a random subset of them as required, rather than obtaining and storing them all. The set of original samples, together with the new samples, describes a function with more particle exchange symmetry or antisymmetry, depending on the choice of sign and compared to the original samples alone. Therefore, when stochastic projection is performed before every time propagation step, it drives the algorithm to converge to a solution with the desired exchange property.

### Regression

As formulated here, regression is a fundamental supervised learning task at which NNs excel. Given an ansatz $\Phi_\vartheta(\mathbf{R})$ and a set of samples $\{(\mathbf{R}_i, \Psi_s(\mathbf{R}_i))\}$ representing our prior knowledge of the true wavefunction at some points in space, we want to find the best value of $\vartheta$. Perhaps the simplest approach is to minimize the sum of squared residuals:

$$J(\vartheta) = \sum_i |\Phi_\vartheta(\mathbf{R}_i) - \Psi_s(\mathbf{R}_i)|^2. \tag{3}$$

To express the ansatz itself, we use a NN. Our architecture is very minimal and has not been tuned for efficiency: the NNs we used consist of a sequence of dense layers with tanh activation functions, and finally a linear output layer. This is in some cases be followed by an optional layer enforcing analytically known boundary and/or cusp conditions

## a

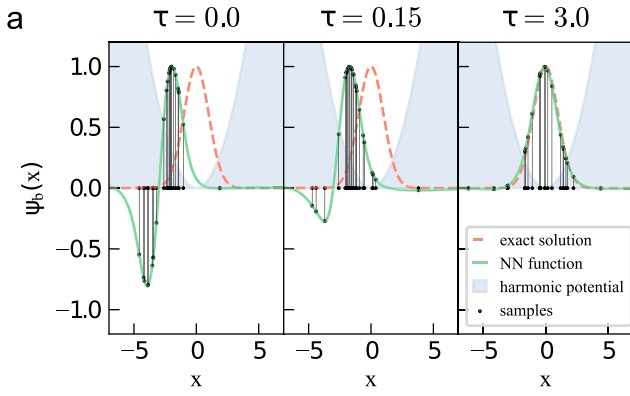

## b

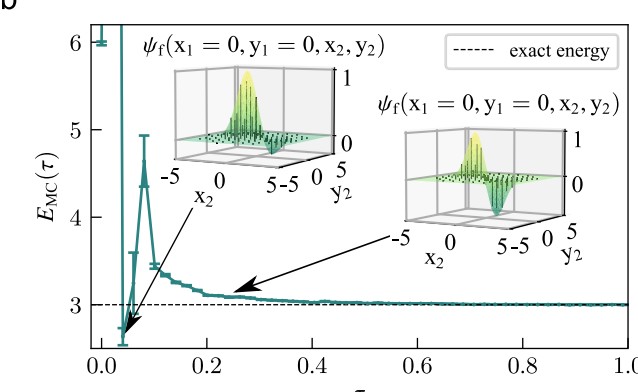

**Fig. 1 | Propagation towards the ground state in a harmonic potential.**
**a** Different steps in the propagation towards the ground state of a particle of mass $m = 1$ in a 1D harmonic oscillator with frequency $\omega = 1$, with $\hbar = 1$. The green line is the function fitted by the neural network to a finite set of samples (black dots on the $x$ axis) and their corresponding values (connected by a black line). Starting with an asymmetric guess ($\tau = 0$), the function converges towards the correct solution (dotted orange line) at the center of the trap and acquires the right symmetry ($\tau = 3$).
**b** Extension of the upper system to two fermions in two spatial dimensions. The energy is estimated by Monte Carlo sampling with error bars showing the standard error, and converges to the ground state value of $E_0 = 3.010 \pm 0.007$, which results in a relative error of 0.35% with respect to the exact value of $E_0^{exact} = 3.0$. The inset shows a cut through the wavefunction. Source data are provided as a Source Data file.

by multiplying the output of the NN with a hand-chosen parameterized function of the coordinates, such as the asymptotic solution at large distances from the origin. Technical details are provided in the Methods section.

The top panel of Fig. 1 shows what this looks like for a single particle in 1D, where the process is easy to visualize. For an arbitrary initial state (top left panel) and the ground state (top right panel) of a 1D harmonic oscillator, a series of samples and the resulting NN-based regression curves are shown. In the insets of the panel below, an analogous visualization is shown for 2D cuts across 4D wavefunctions of two interacting fermions in a 2D harmonic potential.

### Imaginary time propagation

A well-known trick for finding ground states relies on the fact that any wavefunction $|\Psi\rangle$ can be formally described as a superposition $|\Psi\rangle = \sum_\alpha c_\alpha |\Psi_\alpha\rangle$, where $\hat{H}|\Psi_\alpha\rangle = E_\alpha|\Psi_\alpha\rangle$ and $\alpha$ denotes a complete set of quantum numbers. We then have:

$$e^{-\tau\hat{H}}|\Psi\rangle = \sum_\alpha c_\alpha e^{-\tau E_\alpha}|\Psi_\alpha\rangle$$
$$\sim |\Psi_0\rangle + \sum_{\alpha>0}\frac{c_\alpha}{c_0}e^{-\tau(E_\alpha - E_0)}|\Psi_\alpha\rangle. \tag{4}$$

Here, $|\Psi_0\rangle$ and $E_0 < E_{\alpha\neq0}$ denote the ground state (or any state in its degenerate manifold). When $\tau \to \infty$, the last term is exponentially suppressed and—up to a normalization constant—we are left with the ground state. In each iteration of the calculation, marked with the index $j$, we perform a time propagation step at a sampled set of points $\mathbf{R}_i^{(j)}$ over an imaginary time interval of length $\Delta\tau$ (see Eq. (1)). Importantly, we do not actually propagate the ansatz: only a set of samples after propagation $\left\{\left(\mathbf{R}_i^{(j)}, \Psi_s^{(j)}(\mathbf{R}_i)\right)\right\}$ need be obtained. These will be fitted with a new ansatz by regression in the next iteration, after undergoing stochastic projection.

Suppose we are given the trial function from the previous iteration, $\Phi(\mathbf{R},(j-1)\Delta\tau) \equiv \hat{P}_{S/A}\Phi_\vartheta^{(j-1)}(\mathbf{r}_1,\ldots,\mathbf{r}_n)$; and a sample coordinate $\mathbf{R}_i$. If we can obtain the wavefunction $\Phi(\mathbf{R}_i, j\Delta\tau) = \Psi_s^{(j)}(\mathbf{R}_i)$, we will have a sample for the next iteration. By definition, $\Phi(\mathbf{R}_i, j\Delta\tau) = \left[e^{-\Delta\tau\hat{H}}\Phi(\mathbf{R},(j-1)\Delta\tau)\right]\big|_{\mathbf{R}=\mathbf{R}_i}$. For small $\Delta\tau$, this can be approximated by an Euler step:

$$\Phi(\mathbf{R}_i, j\Delta\tau) \simeq \left[\left(1 - \Delta\tau\hat{H}\right)\Phi(\mathbf{R},(j-1)\Delta\tau)\right]\big|_{\mathbf{R}=\mathbf{R}_i}$$
$$= \Phi(\mathbf{R}_i,(j-1)\Delta\tau) \tag{5}$$
$$- \Delta\tau\left[\hat{H}\Phi(\mathbf{R},(j-1)\Delta\tau)\right]\big|_{\mathbf{R}=\mathbf{R}_i}.$$

We therefore need to evaluate the result of applying the Hamiltonian to the trial function at the points $\mathbf{R}_i$, which typically requires taking its Laplacian with respect to the coordinates $\mathbf{R}$. The overall computational cost of this step scales as $O(Nnn_p)$, i.e., linearly with the product of the number of samples $N$; the number of particles $n$; and the number of parameters $n_p$. It is important to note that while the Euler step is convenient and simple, it is by no means a unique choice: for example, one could also evaluate the samples by stochastic path integration, thus avoiding the need to take gradients.

By contrast, in VMC with stochastic reconfiguration time propagation is applied to the ansatz itself, obtaining a new ansatz for $e^{-\tau\hat{H}}\Phi(\mathbf{R},(j-1)\Delta\tau)$ at a cost that scales like the cube of the number of parameters: $O(Nnn_p + n_p^3)$. This, because a matrix of gradients in parameter space needs to be inverted. Storing this matrix also requires presently expensive GPU memory that scales as $O(n_p^2)$, which can also end up being a limiting factor; As we noted earlier, alternatives exist with improved scaling in terms of both memory and computation[43–46]. VMC based on standard gradient descent has an analogous but substantially less expensive update that is defined in the parameter space: $\vartheta \to \vartheta - \eta\nabla_\vartheta\frac{\langle\Phi|\hat{H}|\Phi\rangle}{\langle\Phi|\Phi\rangle}$. This procedure has the same computational scaling as the method proposed here; however, the gradient descent trajectory does not correspond directly to imaginary time propagation and is not as robust at finding the ground state. Furthermore, for fermionic VMC where antisymmetry is enforced by way of determinants in the ansatz, the scaling with the number of particles $n$ becomes cubic; this additional factor is avoided by stochastic projection. We note in passing that the supervised, stochastic approach to natural gradient descent we've introduced here, which scales linearly with $n_p$, may have implications in a much wider domain within machine learning.

### Harmonic oscillator

To investigate how well the method performs in practice, we first consider two particles trapped in a 2D harmonic potential $V(\mathbf{r}) = \frac{1}{2}m\omega^2|\mathbf{r}|^2$. To simplify the various plots, they are presented in units where $\hbar = \omega = m = 1$. Figure 2 shows 2D cuts through the 4D ansatz wavefunctions $\Phi(\mathbf{r}_1,\mathbf{r}_2,\tau)$ at different imaginary times $\tau$ (vertical panels), obtained by setting $y_1 = y_2 = 0$ and plotting the dependence on $x_1$ and $x_2$. The NN has 3 hidden layers with 128 neurons, followed by a linear output layer and a boundary layer that multiplies the output with a Gaussian $e^{-\frac{|\mathbf{r}_1|^2 + |\mathbf{r}_2|^2}{\sigma}}$ having an adjustable width $\sigma$. In all cases, the wavefunctions are normalized such that their maximum absolute value

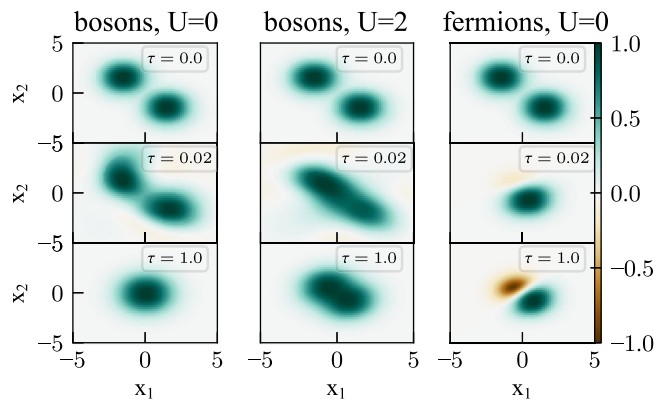

**Fig. 2 | Wavefunction of two particles in a 2D harmonic trap under imaginary time propagation.** In all three cases the initial state ($\tau = 0$) is bosonic. Depending on the projection and interaction, it then converges as imaginary time increases ($\tau = 1.0$) to the ground state of non-interacting bosons (left column), bosons with coulomb interaction $U = 2\hbar\omega$ (middle column) and non-interacting fermions (right column). Source data are provided as a Source Data file.

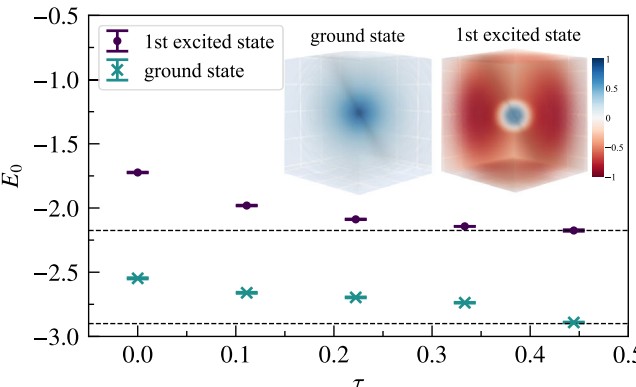

**Fig. 3 | Energies during imaginary time propagation for the (bosonic) ground state and the (fermionic) first excited state of the Helium atom.** Insets show a cut through the wavefunction where one electron is placed at the origin. Error bars indicate the standard error from the Monte Carlo sampling. Source data are provided as a Source Data file.

is 1. To show some nontrivial propagation, the initial guess ($\tau = 0$) for the trial function in all three columns is chosen to be the non-interacting bosonic solution with an offset potential,

$$V(\mathbf{r}) = \frac{1}{2} m\omega^2 \left| \mathbf{r} - \sqrt{\frac{h}{m\omega}} \mathbf{u} \right|^2, \qquad (6)$$

where $\mathbf{u} = (1.5, -1.5)$.

In the left column, we choose a bosonic solution in the stochastic projection, and the particles form a condensate at the center of the trap. In the central column we introduce a repulsive Coulomb interaction $V(\mathbf{r}_1, \mathbf{r}_2) = \frac{U}{|\mathbf{r}_1 - \mathbf{r}_2|}$ into the Hamiltonian; as expected, this suppresses condensation without affecting symmetry. In the right panel we turn off the interaction again, but stochastically project onto the fermionic subspace. Even though we use a bosonic initial guess, the system rapidly converges to the correct antisymmetric solution. This is due to the stochastic projection, which filters out the symmetric component and effectively enhances the antisymmetric one in every propagation step.

The lower panel of Fig. 1 displays the corresponding energies for the fermionic time propagation. During the first few iterations the wavefunction is almost random, and the energy jumps to arbitrary values. Since antisymmetry is not strictly enforced and the bosonic solution always has a lower energy than the fermionic one, the variational theorem does not preclude energies that are too low, and indeed some appear (see left inset and arrow). Eventually, however, the NN learns the correct symmetry and converges exponentially to the exact value $E_0 = 3\hbar\omega$. The standard deviation of the energy, shown here as error bars, provides an additional independent estimator for how close a trial function is to an eigenstate, and also exhibits rapid convergence.

### Helium atom

We finally consider a more realistic model that is difficult to solve by brute-force numerical diagonalization, but still has highly accurate, experimentally verified benchmarks from specialized variational techniques[8,53]. Figure 3 shows the energy for a Helium atom propagated to either its bosonic ground state (purple dots); or its fermionic/triplet first excited state (green dots). The initial condition in both cases is the corresponding non-interacting solution. To avoid numerical issues stemming from the divergence of the Coulomb potential, we follow ref. 54 and multiply the output of the NN by a coulomb cusp function $\Psi_{\text{cusp}}(r_{12}) = c + 2\ln\left(1 + e^{\frac{r_{12}}{a_0}}\right) - \frac{r_{12}}{a_0} - 2\ln(2)$,

where $r_{12} = |\mathbf{r}_1 - \mathbf{r}_2|$. The values obtained after only 4 and 6 time steps, $E_0^{\text{boson}} \approx -2.894 \pm 0.003$ and $E_0^{\text{fermion}} \approx 2.175 \pm 0.006$, are consistent with the benchmarks, shown as dashed horizontal lines. The relative errors with respect to the exact energies are 0.31% for the bosonic and 0.009% for the fermionic state. For the calculation of the ground state we used a NN comprising one hidden layer with 1000 neurons, while for the excited state we used 5 hidden layers with 50 neurons each. The maximum number of samples used was ~$10^6$.

### Discussion

We proposed and tested a machine learning algorithm capable of finding the ground states of both fermionic and bosonic quantum systems. The basic object in our algorithm is a set of samples: random coordinates, and the value of the wavefunction at each coordinate. Unlike VMC, which in ML terms is an unsupervised algorithm, our algorithm is a supervised learning technique built around regression. The ground state is found by imaginary time propagation rather than by directly minimizing the energy of the ansatz through gradient descent; but without the cubic computational scaling in the number of parameters associated with (full) stochastic reconfiguration, or the cubic scaling in the number of particles associated with determinant ansatzes. On the other hand, unlike DMC, there is neither a fixed node approximation nor an uncontrollable sign problem. Stochastic projection allows us to drive the algorithm to learn symmetries rather than explicitly enforcing them on either the ansatz or the initial guess. In principle the model can learn a generic trial function without any guidance, but the method allows for including physical knowledge (such as asymptotics, nodes or Jastrow factors) to accelerate convergence.

We considered a very basic and minimal machine learning model. Future work will explore whether the stochastic representations method can improve the accuracy of VMC with more advanced architectures, explicitly symmetric or otherwise. It would be of great interest to find whether the procedure proposed here can either (a) correctly optimize an ansatz like FermiNet in a case where VMC fails to converge; or (b) show that an unrestricted ansatz can outperform it for some practical problem.

The method presented here can be used on its own, or as an intermediate step in compound VMC/DMC procedures where increasingly complex models are needed in later steps. Its greatest promise is in enabling the application of state-of-the-art, large-scale unrestricted neural models to the challenges of many-body quantum mechanics.

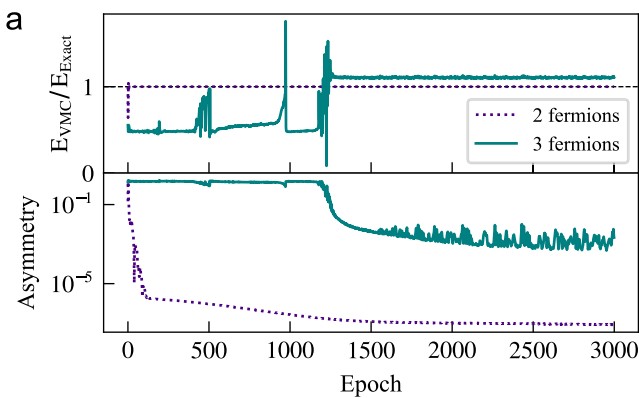

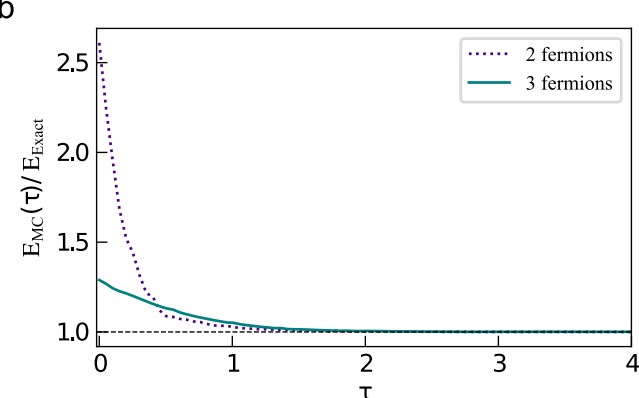

**Fig. 4 | Evolution of the energy towards the ground state of two and three noninteracting fermions in a 1D harmonic potential. a** Performed by variational Monte Carlo with a relative error for two fermions 0.004% and three fermions 10.34% respectively. **b** Using the stochastic representation introduced here with a relative error for two fermions 0.002% and for three fermions 0.004%. The initial guess ($\tau = 0$) for the trial wavefunction is chosen to be the noninteracting fermionic solution with the potential in Eq. (6) and offsets $\mathbf{u} = (-1.0, 1.0)$ for two fermions and $\mathbf{u} = (0.8, 0.0, -0.8)$ for three fermions. Source data are provided as a Source Data file.

## Methods
### Regression parameters
In the examples shown, 80% of samples at iteration $j$ are generated from the probability distribution $|\Phi_\vartheta^{j-1}(\mathbf{R}_i)|^2$ of the previous step (which is the initial guess when $j = 1$) by importance sampling using the Metropolis algorithm. Note that this does not require the wavefunction to be normalized. Another 20% are uniformly distributed in a large domain around the wavefunction, helping to ensure that the wavefunction is suppressed where it should be. The size of the domain in which the samples are distributed is chosen such that it covers all features of the wavefunction. In order to estimate how large that area needs to be, we can consider the (usually analytically tractable) asymptotic behavior and choose a cutoff where the wavefunction is sufficiently suppressed. For all harmonic oscillator cases, we chose a domain where all coordinates are between $-5$ and 5. The restriction of the domain affects the uniformly distributed samples as well as those we generate through importance sampling. Since we obtain the majority of the samples through importance sampling, they will mostly be concentrated in areas where the wavefunction has large absolute values.

All calculations use stochastic gradient decent in the optimization process and a simple network structure with 1–5 hidden layers, each layer consisting of up to 2048 neurons. This proves to be expressive enough even for more complicated wavefunctions. The learning rate is

the hyperparameter we focused on most during optimization. To find its optimal value, we start with a high learning rate and gradually decrease its initial value until the learning curve declines within the first 10 epochs. As a criterion for an accurate fit, we use a mean squared error below $10^{-4}$, which is relatively small compared to the maximal absolute value of the wavefunction (normalized to 1). Our data is divided into a training set and a validation set, and in order to avoid overfitting we always verify that the validation loss does not deviate from the training loss.

### Comparison to VMC
To better judge the effectiveness of our method, we compare our results to calculations performed with VMC. To compare VMC to our technique in a general manner, without resorting to additional ad-hoc assumptions like the use of linear combinations of Slater determinants with Jastrow factors, one can employ dual gradient descent in order to enforce the correct symmetry as a constraint. In dual gradient descent the loss function, which is minimized during training, comprises two terms. One is the energy, and the second includes an asymmetry function $A(\{\mathbf{R}\}, \vartheta)$ that attains its minimal value of 0 when the ansatz respects some symmetry. To enforce fermionic exchange, we chose

$$A(\{\mathbf{R}\}, \vartheta) = \left\langle \frac{\left[\Phi_\vartheta(\mathbf{R}) - (-1)^{\text{sign}(P)} \Phi_\vartheta\left(\hat{P}\mathbf{R}\right)\right]^2}{\left|\{\Phi_\vartheta(\mathbf{R})\}\right|^2} \right\rangle, \tag{7}$$

where in each epoch a random permutation $\hat{P}$ is selected for each sample. When using samples from the distribution $\{\mathbf{R}\} \sim \left\{ |\Phi_\vartheta(\mathbf{R})|^2 \right\}$ the loss has the form

$$J(\vartheta) = \left\langle \frac{\langle \Phi_\vartheta | \hat{H} | \Phi_\vartheta \rangle}{\langle \Phi_\vartheta | \Phi_\vartheta \rangle} \right\rangle + \lambda \cdot A(\{\mathbf{R}\}, \vartheta). \tag{8}$$

We used VMC for two and three noninteracting fermions in a 1D harmonic potential, with a similar network size to that used in our stochastic representation technique. The results are shown in Fig. 4. For two fermions both the symmetry and energy rapidly converge to the correct value and stabilize there, at a computational expense far below what is needed to obtain converged results using the stochastic representation. For three fermions, however, we failed to obtain converged results with VMC. The stochastic representation technique, on the other hand, easily and reliably solves this problem.

### Permutations
Next, we explore the impact of the number of randomly selected permutations on convergence of the energy for two and three fermions in a non-interacting harmonic oscillator in one spatial dimension. The upper panel of Fig. 5 shows that with two fermions, including both permutations for every sample in each time step leads to a correct asymmetric state, while the inclusion of only a single (random) permutation results in convergence to the symmetric state. For three fermions, the lower panel shows that the procedure converges to the fermionic state if a random subset of at least two permutations is chosen for each sample in each step. This suggests that when the data includes enough sample pairs exhibiting exact particle exchange antisymmetry, the model is able to learn this property with a reasonable degree of efficiency.

### Symmetry and scaling
Not explicitly incorporating symmetry or antisymmetry into the ansatz comes at a cost: the need to learn all features of a large permutation group explicitly. We now consider how the number of

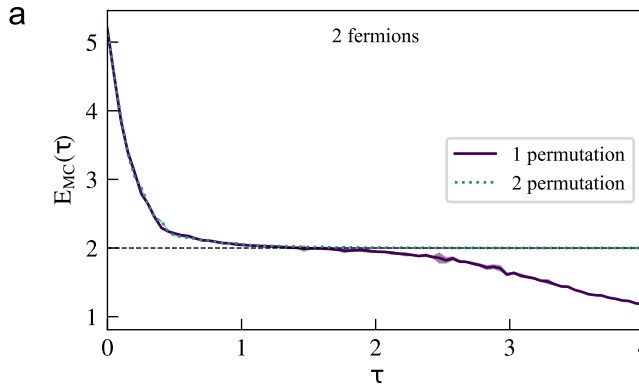

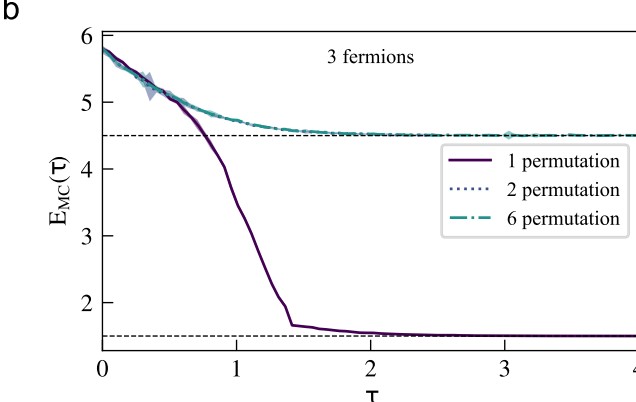

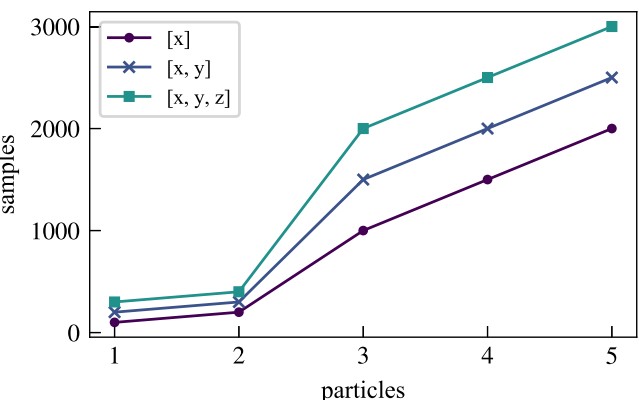

**Fig. 6 | Scaling of the number of samples needed in order to obtain a sufficiently good fit of an $N$-dimensional Slater-determinant.** Source data are provided as a Source Data file.

**Fig. 5 | Energies during imaginary time propagation to the fermionic ground state for non-interacting particles of mass $m = 1$ in a 1D harmonic potential with frequency $\omega = 1$ and $\hbar = 1$.** Different lines refer to the number of randomly selected permutations used in every time step. The shaded regions indicate the standard error from the Monte Carlo sampling. **a** For two fermions including two permutations the final energy has a relative error of 0.005%. **b** For three fermions the final error is 0.004%.

samples scales with the dimension of the wavefunction being fitted, at least for relatively small system sizes. We emphasize that this is a property of the regression alone, and is unrelated to the time propagation or any other aspect of our method. Figure 6 shows how the number of samples needed to obtain a reasonable fit (defined below) scales with the number of noninteracting fermions, for the ground state of the noninteracting harmonic oscillator at various spatial dimensions. We optimized the same neural network to fit a variety of Slater determinants; the different data points differ only in the learning rate, which we tuned for speed. Figure 6 shows that one and two particle wavefunctions are very easy to fit, mostly because the boundary function constituting the last layer in our neural network rather closely approximates the correct solution. For more complex wavefunctions, more samples are needed, but the growth appears approximately linear rather than exponential. For more particles, interactions and more complex potentials, a larger network is eventually needed.

## Implementation
The code is implemented using Google's TensorFlow library[55].

## Data availability
Source data are provided with this paper. The model and wavefunction data are available under restricted access due to their large size and limited utility given that equivalent data can be easily generated from the publicly available code. Access can be obtained by contacting the corresponding author. Source data are provided with this paper.

## Code availability
The source code used to generate this study is open and publicly available in a permanent repository[56].

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

## Acknowledgements

G.C. acknowledges support by the ISRAEL SCIENCE FOUNDATION (Grants No. 2902/21 and No. 218/19) and by the PAZY foundation (Grant No. 318/78).

## Author contributions

H.A. carried out theoretical calculations, developed the code, performed simulations and analyzed the data for the main work on stochastic representation. L.B. carried out theoretical calculations, developed the code, performed simulations and analyzed the data for the scaling and symmetry analysis and for the comparison with variational Monte Carlo. The manuscript was written jointly with input from all authors. G.C. performed early proof-of-concept calculations and supervised the project.

## Competing interests

The authors declare no competing interests.
