## [Peer Review File · Nature Communications]

REVIEWER COMMENTS

Reviewer #1 (Remarks to the Author):

The Manuscript "Stochastic Representation of Many-Body Quantum States" presents a Machine-Learning inspired procedure to obtain a variational approximation of the lowest-energy eigenstate of an Hamiltonian while enforcing a permutational symmetry (or antysymmetry) in the computational basis.

The authors use the procedure to compute the energies of the ground-state and first excited-state of the Helium Atom.

The method presented is valid and will be of interest to the community, and numerical data presented seems correct.

The presentation is overall clearly written.

However, I have two overall negative remarks:

- the analysis on the method the authors propose does not go particularly in depth, leaving several questions unanswered.
- the paper presents weak numerical evidence of the effectiveness of their method: they chose a very simple benchmark problem and avoided more complex ones such as the J1-J2 model (as instead this recent paper [arxiv:2204.07816](https://arxiv.org/abs/2204.07816) does). Overall, it's hard to judge if their method is effective or not because it's not compared to *any* other existing technique. Adding a proper comparison to some data published in an existing article, or by comparing their model to other, existing techniques would make this manuscript a great contribution to the field.

I believe the manuscript should be published after a through revision addressing at least the first point.

Other than the general issue I raised above, below you I list some additional remarks:

- [minor] Introduction, page 1, right column: the authors first state that "[explicit antysymmetric forms] lead to prohibite computational time". I believe that it could help non-domain specific

readers if the authors could be more quantitative on what "prohibite computational time" means. I imagine they refer to factorial growth in this case?

- [wrong] Same paragraph, the authors state that "[the stochastic reconfiguration method] does not scale well to a large number of parameters". Again, the authors should be more specific about what "does not scale well" means. If they refer to the cubic scaling they suggest in the latter part of the manuscript, I believe this to be a wrong statement (see discussion in a later point). This sentence should be revised.

- [minor] I appreciated the succinct schematic overview of the algorithm in the right column of page 2, however the description of step 4 is somewhat unclear. I think the authors could state (assuming I understood correctly) that they are generating a new set of samples " $(R^{\{j\}}, \Psi^{\{j\}})$ ", where the $\Psi^{\{j\}}$ is computed by linearizing the imaginary time propagator starting from the previous samples. they should also state at the end of step 4 that the algorithm loops by going back to step 2.

- [minor] The authors state that the "ansatz can be replaced by a completely different parametrisation as many time as desired...". Since they mention this, but I've never seen this done in practice, I would be curious to know where this could be helpful in practice?

- [major] Page 3, first column: The authors first state that "[P] is one of the $n!$ possible permutation operators", which would suggest that their algorithm would scale factorially-bad as all competitors introduced in the introduction. Instead the authors avoid this problem by "generat[ing] a random subset of [the permutations] as required, rather than obtaining and storing them all."

This is, in my opinion, a central part of their algorithm and the reason why it does not scale factorially. However, an analysis on how to select this "random subset" is missing. I don't doubt that the proposed approach could work, but before recommending this manuscript for publication I believe that this point should be actually studied and analyzed. For example, they could show the relative error of the resulting ground-state and excited-state energy obtained as a function of the way they perform this cut. Crucially, as this grows polynomially, I believe that this should be studied in a few systems where n increases. This study does not necessarily need to be done on a physical system, but could also be carried out on some syntetic dataset.

- [minor] Page 3, right column, "regression": The authors state that "another 20% are uniformly distributed in a large domain around the wavefunction." but don't provide additional details. Could they mention how they selected this 'large domain'?

- [major] Page 3, "regression": The authors don't give details on the stopping criteria of this optimisation, which intuitively I believe might be a very important hyperparameter to the overall performance of the algorithm. If it was hard to make generic statements, they could add a discussion about this to the numerical experiments section (possibly also including the solver they used, like SGD/Adam...)

- [wrong] Page 4, left column: the authors state that SR/NatGrad scales cubically with the number of parameters. However they are not considering the fact that you don't need to invert the Quantum Geometric Tensor matrix, but you only need to solve a linear system which would have a quadratical growth in $\$n_p\$$. Moreover, using some more advanced ways to write the geometric tensor, this

cost can be lowered and the memory cost can be avoided. I discuss those topics on arXiv:2112.10526. Moreover there are recent ML papers (eg arxiv:2202.06236) showing that it's possible to use NatGrad for models with millions of parameters using the same tricks discussed in my paper.

- [minor] Page 5: the authors describe the NN architectures they used for Ground-state and excited-state calculations, but they don't provide motivation as to why they made such choices. Could they comment?

Also, could they give relative errors for their calculations wrt exact values?

I also have some comments about the phrasing of two sentences. Feel free to ignore those as I am not a native english speaker:

- [minor] the authors state "[the DMC breaks down... :] it exhibits a sign problem, noise that grows exponentially with the problem size". While I am not a native english speaker, the phrasing of this sentence sounds off to me. Maybe could be replaced with "it exhibits a sign problem, meaning that the variance of the stochastic estimator (and therefore the error) grows exponentially with the system size."

- [minor] the authors state "this means a dmc calculation should be preceded by...". the authors could rephrase to the midler statement "this suggests that one could improve the precision of the DMC calculation by preceeding it with a variational calculation learning the sign structure", unless they can defend why a variational calculation would be the only way to do that.

Finally, as nature reporting standards require the full disclosure of the source code and protocols upon publication, I would like to suggest to the authors to share their code in a github repository and include a citation to it (possibly through a Zenodo DOI) in the manuscript, in order to better comply with this requirement.

Overall, I thank the authors for writing a clear manuscript that made refereeing it pleasurable, and I hope that my comments can improve the quality of the resulting manuscript

Sincerely,

Filippo Vicentini

Reviewer #2 (Remarks to the Author):

The paper "Stochastic Representation of Many-Body Quantum States" presents a very interesting combined sampling and learning approach to approximate states of the stationary many-body Schrödinger equation. The basic idea is to do imaginary time propagation as in diffusion Monte Carlo (DMC), but to then sample particle permutations and enforce symmetry or antisymmetry and subsequently approximate the symmetric or antisymmetric wavefunction with a ML regression model. Due to this combination the authors demonstrate that they can approximate both bosonic and fermionic wavefunctions without being restricted by a fixed nodal surface as in DMC.

I really like this idea and think it is a very creative and original work that cleverly combines quantum Chemistry and machine learning ideas and the article is potentially interesting for the readership of Nat Comms.

My main worry is the scalability of the machine learning component of the approach. Specifically, in the regression step it seems that the aim is to approximate the wave function globally. Since the ML model doesn't incorporate symmetry or antisymmetry explicitly, it needs to learn all features of the 3N-dimensional wavefunction explicitly. The authors generate random particle permutations in every iteration, but since there are exponentially many perturbations this basically corresponds to training a symmetric (or antisymmetric) function with data augmentation. This will not scale to an exponentially large symmetry group. I agree that not explicitly building in symmetries in the neural net simplifies the ML structure and increases expressiveness - but this does not come without cost in terms of learning efficiency. Perhaps I have missed some important point, in that case I'd invite the authors to explain.

Despite this point I feel that the method goes in a very interesting new direction, so I don't want to dismiss the paper based on it. To prove the point that the method can scale to larger and more complex systems, I would suggest to demonstrate it on a higher-dimensional system that would suffer from a poor global approximation of the wavefunction. An example is H10 chain in the stretched configuration.

A loosely related paper that also employs the dynamic formulation of the electronic SE in order to find stationary states with kernel regression is this one, although the ML representation is quite different perhaps this is useful for the authors:

<https://iopscience.iop.org/article/10.1088/2632-2153/ac14ad/meta>

There are a few minor formatting issues, e.g. the Tensorflow citation is incomplete.

I'm happy to reconsider a revised version.

Reviewer #1

The Manuscript "Stochastic Representation of Many-Body Quantum States" presents a Machine-Learning inspired procedure to obtain a variational approximation of the lowest-energy eigenstate of an Hamiltonian while enforcing a permutational symmetry (or antysymmetry) in the computational basis. The authors use the procedure to compute the energies of the ground-state and first excited-state of the Helium Atom.

The method presented is valid and will be of interest to the community, and numerical data presented seems correct. The presentation is overall clearly written.

We thank the referee for this brief and accurate summary, as well as the positive assessment.

However, I have two overall negative remarks:

- the analysis on the method the authors propose does not go particularly in depth, leaving several questions unanswered.

We thank the referee for this comment; our revision now includes a much deeper analysis that answers all these questions; please see more about this below.

*- the paper presents weak numerical evidence of the effectiveness of their method: they chose a very simple benchmark problem and avoided more complex ones such as the J1-J2 model (as instead this recent paper [arxiv:2204.07816](https://arxiv.org/abs/2204.07816) does). Overall, it's hard to judge if their method is effective or not because it's not compared to *any* other existing technique. Adding a proper comparison to some data published in an existing article, or by comparing their model to other, existing techniques would make this manuscript a great contribution to the field.*

This is a great point and we thank the referee for raising it. The manuscript was originally a short letter introducing the idea and its potential, without making any claims about absolute efficiency compared to existing methods. This, because such comparisons are generally complicated by details like the quality of implementation and hardware availability, and we assumed they would be of interest chiefly to a more specialized audience.

This comment, as well as one by the second referee, convinced us to revise our opinion and include a comparison to an existing technique. We believe this greatly strengthens our manuscript.

The referee suggested a comparison to spin models. This is an intriguing direction and the newly added references [45–46] from the Feiguin group, which propose an approximate approach to natural gradient descent for J1-J2 models, suggests that it would be highly productive. However, it lies beyond the scope of the current work, which focuses on real-space Hamiltonians; with respect, we note that these continuous problems are in practice more difficult to solve. Here, our approach also has the additional advantage over most methods that no orbital basis need be defined and converged or matrix elements evaluated; we work directly in an infinite-dimensional, continuous Hilbert space. Nevertheless, a future study considering the stochastic representation approach to phenomenological lattice Hamiltonians—either of the bosonic, fermionic or spin type—could be the basis of an extremely interesting future research project, where indeed high quality benchmarks are available.

Figure 1: Evolution of the energy towards the ground state of two and three noninteracting fermions in a harmonic potential, performed by VMC (left) and our method (right).

The most direct comparison we could make was to calculations performed with Variational Monte Carlo (VMC). To compare VMC with our technique in a general manner, without resorting to additional ad-hoc assumptions like the use of linear combinations of Slater determinants with Jastrow factors, one can employ dual gradient descent in order to enforce the correct symmetry as a constraint. In dual gradient descent the loss function, which is minimized during training, comprises two terms. One is the energy, and the second includes a newly defined ‘‘Asymmetry’’ function $A(\{\mathbf{R}\}, \vartheta)$ that attains its minimal value of 0 when the ansatz respects a symmetry. To enforce fermionic exchange, we chose

$$A(\{\mathbf{R}\}, \vartheta) = \left\langle \left(\frac{\{\Phi_{\vartheta}(\mathbf{R})\} - \{(-1)^{\text{sign}(P)} \Phi_{\vartheta}(\hat{P}\mathbf{R})\}}{|\{\Phi_{\vartheta}(\mathbf{R})\}|} \right)^2 \right\rangle,$$

where in each epoch a random permutation \hat{P} is performed for each sample. When using samples from the distribution $\{\mathbf{R}\} \sim \{|\Phi_{\vartheta}(\mathbf{R})|^2\}$ the loss has the form

$$J(\vartheta) = \left\langle \frac{\langle \Phi_{\vartheta} | \hat{H} | \Phi_{\vartheta} \rangle}{\langle \Phi_{\vartheta} | \Phi_{\vartheta} \rangle} \right\rangle + \lambda \cdot A(\{\mathbf{R}\}, \vartheta).$$

The Lagrange multiplier λ is determined by the dual gradient descent procedure.

We performed VMC for 2 and 3 noninteracting fermions in a 1D harmonic potential, with a similar network size to that used in our stochastic representation technique. The results are shown in Fig. 1. For 2 fermions both the symmetry and energy rapidly converge to the correct value and stabilize there, at a computational expense far below what is needed to obtain converged results using the stochastic representation. For 3 fermions, however, we failed to obtain converged results with VMC. The stochastic representation technique, on the other hand, easily and reliably solves this problem.

I believe the manuscript should be published after a through revision addressing at least the first point.

As we now clearly showcase a simple problem where our method easily overcomes a hard failure of the standard approach, we believe that the referee’s main concern has been fully addressed.

Other than the general issue I raised above, below you I list some additional remarks:

- [minor] Introduction, page 1, right column: the authors first state that ‘‘[explicit antisymmetric forms] lead to prohibite computational time’’. I believe that it could help non-domain specific readers if the authors could be more quantitative on what ‘‘prohibite computational time’’ means. I imagine they refer to factorial growth in this case?

We indeed refer to the fact that most quantum chemistry methods rely on representing electronic wavefunctions as linear combinations of determinants. Since the number of relevant determinants grows exponentially with system size (recent methods like PauliNet and FermiNet being a potential exception to this) and the cost of evaluating each determinant grows cubically, the computational cost of high-accuracy methods increases rapidly with the number of electrons. We now note this in the manuscript and are thankful for the suggestion.

- [wrong] Same paragraph, the authors state that "[the stochastic reconfiguration method] does not scale well to a large number of parameters". Again, the authors should be more specific about what "does not scale well" means. If they refer to the cubic scaling they suggest in the latter part of the manuscript, I believe this to be a wrong statement (see discussion in a later point). This sentence should be revised.

We thank the referee for pointing out this vague phrasing and the interesting recent advances in this field. We have amended the sentence and included the references suggested in the later point.

- [minor] I appreciated the succinct schematic overview of the algorithm in the right column of page 2, however the description of step 4 is somewhat unclear. I think the authors could state (assuming I understood correctly) that they are generating a new set of samples " $(R^{\{j\}}, \Psi^{\{j\}})$ ", where the $\Psi^{\{j\}}$ is computed by linearizing the imaginary time propagator starting from the previous samples. they should also state at the end of step 4 that the algorithm loops by going back to step 2.

We appreciate this helpful remark and apologize for the lack of clarity. The relevant text has been cleaned up and should now be easier to understand, including making it more straightforward to see that the last three steps are iterated. We still prefer to defer the specific way in which imaginary time propagation is performed to later in the paper: the linearization, while used in all the results shown in our manuscript, is just one possible choice of algorithm for this purpose.

- [minor] The authors state that the "ansatz can be replaced by a completely different parametrisation as many time as desired...". Since they mention this, but I've never seen this done in practice, I would be curious to know where this could be helpful in practice?

Here, we refer to the flexibility of our method. At every point of the time propagation we have the option to change the network's architecture, learning parameters and sample generation strategy. This can in fact be very useful, when starting with an ansatz that differs greatly from the ground state wavefunction the algorithm will eventually converge to. The model required to obtain a good fit of the wavefunction can change as the wavefunction evolves during time propagation, e.g. from a Gutzwiller ansatz to a simple network, and then eventually to a larger one for high accuracy. In the examples presented in the manuscript, however, we did not in practice change the network structure during propagation, except in the sense of starting from an initial guess that is not given by a network at all. We now explain this in the text.

- [major] Page 3, first column: The authors first state that "[P] is one of the $n!$ possible permutation operators", which would suggest that their algorithm would scale factorially-bad as all competitors introduced in the introduction. Instead the authors avoid this problem by "generat[ing] a random subset of [the permutations] as required, rather than obtaining and storing them all." This is, in my opinion, a central part of their algorithm and the reason why it does not scale factorially. However, an analysis on how to select this "random subset" is missing. I don't doubt that the proposed approach could work, but before recommending this manuscript for publication I believe that this point should be actually studied and analyzed. For example, they could show the relative error of the resulting ground-state and excited-state energy obtained as a function of the way they perform this cut. Crucially, as this grows polynomially, I believe that this should be studied in a few systems where n increases. This study does not necessarily need to be done on a physical system, but could also be carried out on some syntetic dataset.

Figure 2: Energies during imaginary time propagation to the fermionic ground state for two (left panel) and three (right panel) noninteracting particles of mass $m = 1$ in a 1D harmonic potential with frequency $\omega = 1$ and $\hbar = 1$. Different lines refer to the number of randomly selected permutations used in every time step.

We thank the referee for this insightful remark. First, we note that our method could in principle be used with an explicitly antisymmetric ansatz, thereby completely bypassing this issue. The ability to stochastically symmetrize unrestricted ansatzes is an advantage, not a limitation, of the stochastic representation of wavefunctions.

Nevertheless, the referee raises an interesting question and we have indeed done as suggested, by exploring the impact of the number of randomly selected permutations on the convergence of the energy for 2 and 3 noninteracting fermions in a 1D harmonic potential. Because the number of possible permutations rises from 2 to 6 in this case, this analysis is of minimal but sufficient complexity to answer the question raised by the referee. The left panel of Fig. 2 shows that with two fermions, including both permutations for every sample in each time step leads to a correct asymmetric state, while the inclusion of only a single (random) permutation results in convergence to the symmetric state. For three fermions, the right panel shows that the procedure converges to the fermionic state if a random subset of at least two permutations is chosen for each sample in each step. This suggests that when the data includes enough sample pairs exhibiting exact particle exchange antisymmetry, the model is able to learn this property with a reasonable degree of efficiency. This analysis is now included in the manuscript.

For an additional way of investigating this scaling, we also refer the referee to Fig. 3 and its discussion below.

- [minor] Page 3, right column, "regression": The authors state that "another 20% are uniformly distributed in a large domain around the wavefunction." but don't provide additional details. Could they mention how they selected this 'large domain'?

We thank the referee for pointing out this oversight, and the information is now provided both here and in the manuscript itself. The size of the domain in which the samples are distributed is chosen such that it covers all features of the wavefunction. In order to estimate how large that area needs to be, we can consider the (usually analytically tractable) asymptotic behavior and choose a cutoff where the wavefunction is sufficiently suppressed. The restriction of the domain affects the uniformly distributed samples as well as those we generate through importance sampling. For all wavefunctions in the manuscript a domain where all coordinates are between -5 and 5 was chosen. Since we obtain the majority of the samples through importance sampling, they will mostly be concentrated in regions where the wavefunction has large absolute values.

- [major] Page 3, "regression": The authors don't give details on the stopping criteria of this optimization, which intuitively I believe might be a very important hyperparameter to the overall performance

of the algorithm. If it was hard to make generic statements, they could add a discussion about this to the numerical experiments section (possibly also including the solver they used, like SGD/Adam...)

We apologize for the lack of details regarding the optimization process. In all calculations, we used stochastic gradient descent for the optimization. As a criterion for a sufficiently good fit, we chose a mean squared error below 10^{-4} . This is a relatively small value compared to the maximum absolute value of the wavefunction, which is always normalized to 1.0; because all actual learning is supervised, this is relatively straightforward. 33% of our data is randomly chosen for validation and we always make sure that the validation error does not deviate from the training error to avoid overfitting. These important details are now included in the manuscript.

- [wrong] Page 4, left column: *the authors state that SR/NatGrad scales cubically with the number of parameters. However they are not considering the fact that you don't need to invert the Quantum Geometric Tensor matrix, but you only need to solve a linear system which would have a quadratical growth in n_p . Moreover, using some more advanced ways to write the geometric tensor, this cost can be lowered and the memory cost can be avoided. I discuss those topics on arXiv:2112.10526. Moreover there are recent ML papers (eg arxiv:2202.06236) showing that it's possible to use NatGrad for models with millions of parameters using the same tricks discussed in my paper.*

We thank the referee for pointing out these advances, which we now mention. We changed the reference to the computational scaling, but note that the approaches described in these papers rely on QR decomposition to solve the linear system. The exact evaluation of this still scales cubically. Nevertheless, further approximations can be made and indeed memory, which is a crucial bottleneck, can be saved.

- [minor] Page 5: *the authors describe the NN architectures they used for Ground-state and excited-state calculations, but they don't provide motivation as to why they made such choices. Could they comment?*

A simple dense network structure consisting of between 1 and 5 hidden layers with up to 2048 neurons per layer was used in all cases. This, because it is the simplest architecture we could imagine that worked: our choice was motivated entirely by the desire for minimal conceptual complexity. We can imagine that more optimal architectures exists, but have yet to investigate this.

Also, could they give relative errors for their calculations wrt exact values?

This is a good suggestion, and we have added relative errors for all cases to the manuscript.

I also have some comments about the phrasing of two sentences. Feel free to ignore those as I am not a native english speaker:

- [minor] *the authors state "[the DMC breaks down... :] it exhibits a sign problem, noise that grows exponentially with the problem size". While I am not a native english speaker, the phrasing of this sentence sounds off to me. Maybe could be replaced with "it exhibits a sign problem, meaning that the variance of the stochastic estimator (and therefore the error) grows exponentially with the system size."*

- [minor] *the authors state "this means a dmc calculation should be preceded by...". the authors could rephrase to the midler statement "this suggests that one could improve the precision of the DMC calculation by preceding it with a variational calculation learning the sign structure", unless they can defend why a variational calculation would be the only way to do that.*

We agree with the referee and have rephrased these sentences.

Finally, as nature reporting standards require the full disclosure of the source code and protocols upon publication, I would like to suggest to the authors to share their code in a github repository and include a citation to it (possibly through a Zenodo DOI) in the manuscript, in order to better comply with this requirement.

We thank the referee for this great suggestion, which we liked very much. A public repository containing our code is now provided and archived on Zenodo. We also included instructions on how to use it, in an attempt to make our results as easy to reproduce as possible.

Overall, I thank the authors for writing a clear manuscript that made refereeing it pleasurable, and I hope that my comments can improve the quality of the resulting manuscript

Indeed they have!

Reviewer #2

The paper "Stochastic Representation of Many-Body Quantum States" presents a very interesting combined sampling and learning approach to approximate states of the stationary many-body Schrödinger equation. The basic idea is to do imaginary time propagation as in diffusion Monte Carlo (DMC), but to then sample particle permutations and enforce symmetry or antisymmetry and subsequently approximate the symmetric or antisymmetric wavefunction with a ML regression model. Due to this combination the authors demonstrate that they can approximate both bosonic and fermionic wavefunctions without being restricted by a fixed nodal surface as in DMC.

I really like this idea and think it is a very creative and original work that cleverly combines quantum Chemistry and machine learning ideas and the article is potentially interesting for the readership of Nat Comms.

We thank the referee for the positive remarks and for this summary, which neatly and concisely captures the main ideas we presented.

My main worry is the scalability of the machine learning component of the approach. Specifically, in the regression step it seems that the aim is to approximate the wave function globally. Since the ML model doesn't incorporate symmetry or antisymmetry explicitly, it needs to learn all features of the $3N$ -dimensional wavefunction explicitly. The authors generate random particle permutations in every iteration, but since there are exponentially many perturbations this basically corresponds to training a symmetric (or antisymmetric) function with data augmentation. This will not scale to an exponentially large symmetry group. I agree that not explicitly building in symmetries in the neural net simplifies the ML structure and increases expressiveness - but this does not come without cost in terms of learning efficiency. Perhaps I have missed some important point, in that case I'd invite the authors to explain.

This is a great comment, and we agree with the referee that this is a well-justified concern that requires further investigation. In fact, it is a special case of a larger issue that is ubiquitous in machine learning: there is often a trade off between the increased expressiveness of unrestricted models, and the guaranteed compliance with exactly known properties of restricted ones. At very large scale, it appears that the former approach often prevails, but this is by no means to be taken as the general state of things. In the present context of particle exchange symmetry, many types of models with and without explicit constraints have been considered in the literature (see e.g. arXiv:2007.15298), but no clear winner has yet emerged.

We also note, as we did with the previous referee, that our method could in principle be used with an explicitly antisymmetric ansatz, thereby completely bypassing this issue. The ability to stochastically symmetrize unrestricted ansätze is an advantage, not a limitation, of the stochastic representation of wavefunctions.

In practice, it is straightforward—if perhaps somewhat surprising—to verify that with the simple ansatz used here, the number of samples needed to obtain a good fit shows no evidence of scaling exponentially with the number of particles. This, at least for the relatively small system size we've studied so far. We emphasize that this is a property of the regression alone, and is unrelated to the time propagation or any other

Figure 3: The number of samples needed to fit the noninteracting ground state of different numbers of particles in a harmonic potential, for different physical dimensions.

aspect of our method. Therefore, in Fig. 3, we investigated how the number of samples needed to obtain a reasonable fit scales with the number of noninteracting fermions in the ground state of the noninteracting harmonic oscillator, at various spatial dimensions. Thus, we fit the same neural network to a variety of Slater determinants; the different data points vary only in the learning rate, which we tuned for speed. The figure below shows that one and two particle wavefunctions are very easy to fit, mostly because the boundary function constituting the last layer in our neural network rather closely approximates the correct solution. For more complex wavefunctions, more samples are needed, but the growth appears approximately linear rather than exponential. For more particles, interactions and more complex potentials, a larger network is eventually needed.

This analysis is now included in the manuscript.

Despite this point I feel that the method goes in a very interesting new direction, so I don't want to dismiss the paper based on it. To prove the point that the method can scale to larger and more complex systems, I would suggest to demonstrate it on a higher-dimensional system that would suffer from a poor global approximation of the wavefunction. An example is H10 chain in the stretched configuration.

The H10 chain is a very useful and important benchmark that we will certainly address in a future study. However, within the context of the present manuscript, treating it would be both very expensive, and—more importantly—beside the point. The former, because (as we also mentioned in a reply to the other referee) to keep things conceptually simple, we are using an extremely naive model architecture that most certainly should not be expected to scale well to large systems: a dense network with a few layers. The latter, because it has already been shown that with, e.g., the FermiNet or PauliNet architectures, it is possible to solve this problem to a quite reasonable level of accuracy using traditional VMC. The main focus of our manuscript is an alternative to VMC, not an alternative to FermiNet; one of its advantages is that it allows for models which, unlike FermiNet, are not explicitly antisymmetric.

It will be interesting, in the future, to check whether or not our procedure can improve on the accuracy of VMC for, e.g., H10 with FermiNet or PauliNet; in both cases the stochastic symmetrization will be unnecessary, because the ansatzes are restricted. It would be much more interesting, however, to find whether our procedure can either (a) correctly optimize an ansatz like FermiNet in a case where VMC fails, as we demonstrated for a naive ansatz and a simple state in Fig. 1; or (b) show that an unrestricted ansatz can outperform FermiNet for some practical problem. These challenges require significant research in new directions, and are therefore outside the scope of the present manuscript, which intends only to present the stochastic representation of wavefunctions.

We have clarified these points in the manuscript, making sure to emphasize that we are not claiming that

the model architecture used here is competitive with state-of-the-art models in the literature, regardless of whether it is employed in the stochastic representation method.

A loosely related paper that also employs the dynamic formulation of the electronic SE in order to find stationary states with kernel regression is this one, although the ML representation is quite different perhaps this is useful for the authors: <https://iopscience.iop.org/article/10.1088/2632-2153/ac14ad/meta>

We thank the referee for pointing us to this relevant work, which we now cite. Indeed, our main ideas are independent of the representation, and kernel regression may well be more efficient than neural networks in small-dimensional cases.

There are a few minor formatting issues, e.g. the Tensorflow citation is incomplete.

We thank the referee for spotting this formatting error, and have corrected it.

I'm happy to reconsider a revised version.

We are grateful for the referees' comments and critique, which have not only helped us correct several mistakes and misprints, but also significantly improved the readability of the manuscript and deepened the analysis of our method. Since we addressed the two major concerns above and further strengthened our manuscript, we hope it can now be accepted for publication in Nature Communications.

REVIEWERS' COMMENTS

Reviewer #1 (Remarks to the Author):

I believe that the authors have addressed in a satisfactory manner all the issues I raised in my original report, and believe that after the revisions, the manuscript in the current state will be of interest to the readers of Nature communications and should be published.

The method is sound, the presentation is clear, and there is now a comprehensive enough analysis of the method.

The work is original enough and deviates from existing approaches to warrant publication in this journal.

While I have slight disagreements with a few statements that the authors made in their paper, I think that the discussion there is quite in depth and would hope the authors will publish it along with the paper.